# Effect of Crack Orientation on Fatigue Life of Reinforced Concrete Bridge Decks

**Eissa Fathalla [1]** **, Yasushi Tanaka [2] and Koichi Maekawa [3,\***

[1]  Department of Civil Engineering, The University of Tokyo, 7-3-1 Hongo, Bunkyo-ku, Tokyo 113-8656, Japan; eissa.tokyo.concrete@gmail.com

[2]  Department of Civil and Environmental Engineering, Kanazawa Institute of Technology, 7-1 Ohgigaoka, Nonoichi, Ishikawa 921-8501, Japan; ytanaka@neptune.kanazawa-it.ac.jp

[3]  Department of Civil Engineering, Graduate School of Urban Innovation, Yokohama National University, 79-1 Tokiwadai, Hodogaya, Yokohama 240-8501, Japan

\*  Correspondence: maekawa-koichi-tn@ynu.ac.jp; Tel.: +81-45-339-4155

**Abstract:** In visual inspection of bridges at sites, much attention is given to the density and width of cracks of concrete, but little attention is paid to crack orientation for the diagnosis of bridge performance. In this research, the effect of crack orientation on the remaining fatigue life of reinforced concrete (RC) bridge decks is investigated for crack patterns with a wide range of possible crack orientations. The data assimilation technology of multi-scale simulation and the pseudo-cracking method, which are widely validated for fatigue-lifetime simulation, are utilized in this study. The impact of the crack direction on fatigue life is found to be associated with the coupled flexure-shear mode of failure, and the mechanism to arrest shear cracking by preceding cracks is quantitatively estimated. This mechanism is similar to the stop-hole to prevent fatigue cracks in steel structures, and it enables us to enhance the fatigue life of RC decks. It is demonstrated that the crack orientations that approximate the longitudinal and transverse directions of RC decks are the ones that most extend remaining fatigue life. Finally, the higher risk cracking locations on the bottom surface of RC decks are discussed, presenting information of use to site inspectors.

**Keywords:** fatigue loading; bridge decks; pseudo-cracking method; data assimilation

## 1. Introduction

Bridges are essential infrastructure for transportation and trade. However, throughout the world, fatigue-inducing repetitive traffic loads and environment-related defects, such as corrosion, thermal and shrinkage cracking, and freezing and thawing, lead to degradation of reinforced concrete (RC) bridge decks [1–9]. Particularly in areas of higher seismic activity, where the reduced weight of bridge viaducts is beneficial, the fatigue damage of thinner RC decks whose thickness is less than 200 mm is a common issue. Moreover, the capacity of concrete declines significantly compared to its ultimate short-term strength [10–16]. In Japan, the preservation plan for Japanese highways [17] calls for more than 50% of the total maintenance budget to be spent on the renewal and repair of old RC decks, which were designed and constructed before the 1970s. Thus, the remaining fatigue life of RC decks needs to be rationally estimated for reliable social stock management.

When assessing the health of bridges, inspectors direct their attention to the density and width of the cracks appearing on the concrete surfaces of RC decks, as shown in Figure 1. RC is designed to allow cracking under service loads, and the orientation of cracks in RC is related to its performance [18–22]. The fact that crack-to-crack interactions may cause behavioral changes in structural members has been proven, and this has been investigated for beams and panels, where crack propagation can be observed

by the naked eye. In fact, crack interactions have been investigated in view of the serviceability and ultimate limit states [20–22].

In the case of bridge decks, however, it is hard to detect crack interactions since cracks develop out of view inside the structure. Thus, it is of practical importance to apply knowledge of structural mechanics to site inspections in terms of crack orientation and crack-to-crack interactions. This study focuses on the effect of crack orientation on the remaining fatigue life of RC decks with consideration of a wide range of crack orientations. The core technologies employed in this study are multi-scale cracked concrete simulation [23,24] and the pseudo-cracking method [25,26], which have been validated to effectively assess the remaining fatigue life of RC decks [20–22,27].

First, the authors built a dataset of various crack patterns to study bridge performance by using a randomized artificial crack pattern program (RACP) to overcome the biased real cracks at sites and cover the entire range of crack orientations. The RACP was made to produce fictitious but probable cracks and was applied to build the training datasets of neural network models [21,22]. Crack density and width were kept fixed in the scheme of the RACP, and only crack direction was changed in this study. The effect of crack orientation on the remaining fatigue life of RC decks was discussed, and crack orientation was found to be associated with the coupled flexure-shear action, where pre-cracks tend to arrest the propagation of post-shear cracks. Finally, after capturing the general trend of the crack orientation effect on fatigue life, the impact of crack location was studied again in accordance with the most sensitive crack orientations.

The pre-crack arrest mechanism in RC members closely approximates that of the stop-holes that are drilled in steel members to stop the growth of fatigue cracks [28–31]. The pre-cracks in RC members act as stop-holes, stopping the propagation of post-shear cracks and resulting in extended fatigue life. Thus, this arrest mechanism enables us to upgrade the fatigue life of RC decks and to conduct a fair assessment of RC decks that may present a rough appearance, but have mechanically enhanced fatigue life.

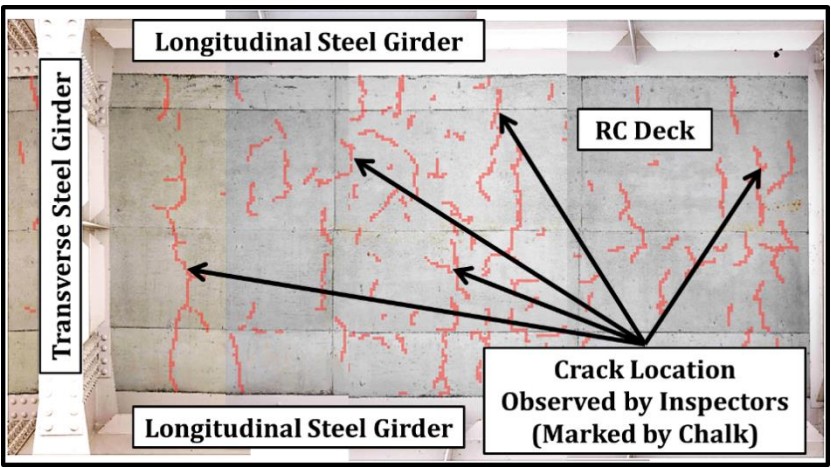

**Figure 1.** Site inspection for bottom surface cracks of reinforced concrete (RC) decks.

## 2. Methodology and Studied Cases

### 2.1. Data Assimilation Technology

For fatigue-lifetime simulation of RC decks based on site inspected cracks, the data assimilation technology of the multi-scale simulation program and the pseudo cracking method, which are broadly validated [23–26], were utilized. The constitutive laws of the multi-scale simulation were upgraded in the last few decades to deal with high cycle fatigue, so they can account for fatigue damage of concrete by a decrease in both strength and stiffness, and increase in time dependent deformations on the basis of the direct path integral method [24]. As concrete is a cementitious composite, a single

crack opening (Mode-I) though cement paste solid exhibits tension softening and the fracture energy is much less than that of crack shear transfer of Mode-II (see Figure 2) owing to the interlocking of rough crack surfaces. Then, blunt multiple cracks are consequently dispersed, and the direction of subsequently propagating cracks is assumed to coincide with that of the updated principal stress, which satisfies the equilibrium with non-orthogonal preceding cracks. It means the minimum fracture energy of the whole analysis domain associated with the shear transfer along crack planes and tension softening normal to cracks [24]. The high shear transfer along preceding crack planes of concrete is a fatigue resistant mechanism which differs from the case of smooth preceding crack planes without shear transfer [32].

Generally, the existing crack will propagate if its propagation is associated with a decrease in the total energy of the system [33,34]. In the case of RC members, the post-shear cracks propagate through the preceding cracks when the shear can transfer from one crack's plane to another. Therefore, the system may easily reach the point of minimum potential energy. On the other hand, when the preceding crack's arrest mechanism occurs, the system is in a stable state, however, more energy is required to create independent shear cracks until forming a failure path. These are so-called cracks interactions, which are explained in detail in later sections.

On the other hand, the pseudo cracking method [26] is a numerical technique for fatigue life of RC decks based upon site-inspected cracks on the surfaces of RC decks. Internal unknown cracks are generated in the finite element model in the early cycles of fatigue loading by the corrector-predictor approach based on energy principles in order to satisfy dynamic equilibrium and deformational compatibility [26].

The multi-scale simulation program can successfully capture the response of a multi-cracked element under multi-stress conditions [18,23], where the global response is an assembly of local crack responses of all the cracks inside the element. The loading conditions of each crack must satisfy the deformational compatibility and equilibrium in its local direction. Through this process, some cracks are activated, while others are idle, resulting in a mechanism called anisotropic crack interaction. The activated cracks dominate the overall behavior, while other cracks are dormant. The activation of pre-cracks reduces the stress concentration in the diagonal direction, where it may affect the response of new cracks. Thus, crack interactions (activation or dormancy) may govern the global structural behavior of RC members. It should be noted that the kinematics of cracks of concrete including opening/closing, slip, and anisotropic interactions are already integrated into the multi-scale simulation program [18,23].

Figure 3 shows examples of the validation of data assimilation technology. Figure 3a shows an actual RC deck from a site with current damage [26] that was tested experimentally to check its remaining fatigue life. Next, it was analyzed by the data assimilation technology for validation. It is clear that the simulation results are in good agreement with the experimental ones. Figure 3b shows another example of validation for an experimental specimen [25], where the data assimilation technology succeeded to predict its remaining fatigue life at different time intervals of damage.

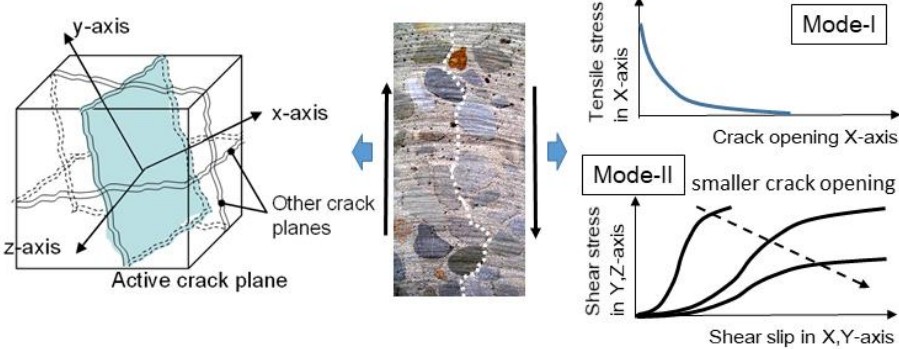

**Figure 2.** Tension and shear stress transfer across crack planes of concrete.

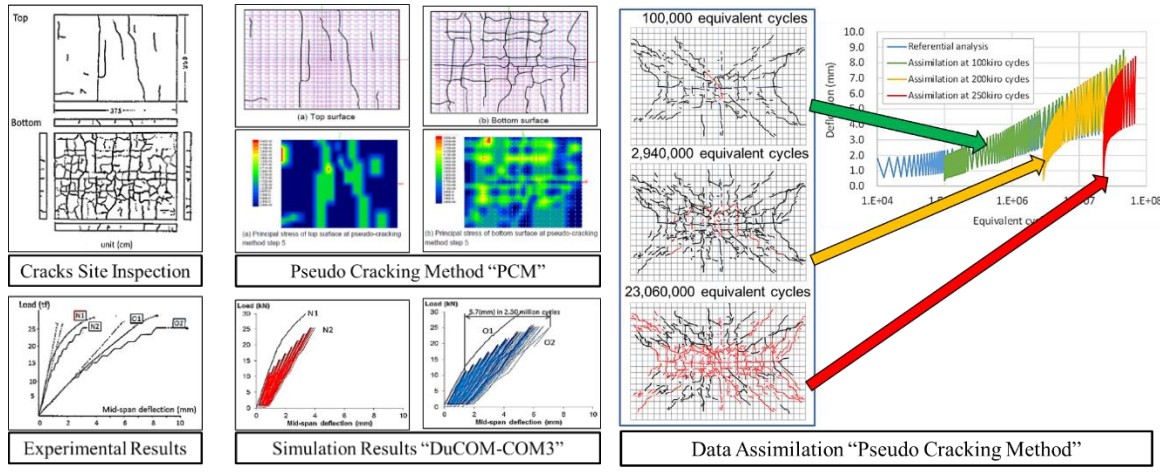

(**a**) Actual RC deck from site.      (**b**) Experimental specimen.

**Figure 3.** Validations of data assimilation technology.

*2.2. Studied RC Deck*

The RC decks of road steel-girder bridges are targeted in this study. Such decks are vertically supported by main steel girders in the longitudinal direction (traffic direction), as shown in Figure 4. These main girders are connected laterally by transverse girders to secure their lateral stability. Generally, RC decks are subjected to a heavy regimen of repeated traffic loads as the result of daily use, leading to deterioration, as stated in the introduction section.

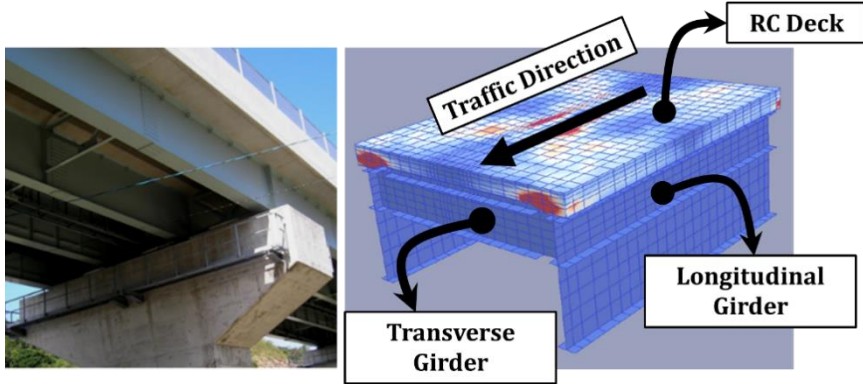

**Figure 4.** RC deck of road steel-girder bridges.

2.2.1. Dimensions and Reinforcement

Here, let us focus on fatigue loading in the dry state [20,21]. The targeted deck slab is a thin RC plate of a type that was typically built in the past and is still in service. Figure 5 shows the dimensions and the arrangement of reinforcing bars. Generally, bridge decks follow the conceptual design of one-way slabs supported by longitudinal girders, while the length of RC decks depends on several aspects such as the type of bridge. In a previous study [20], the authors chose the length of 6.0 m as the optimum length for fulfilling engineering practices at sites and effective analysis of RC decks, and thus, this length was also selected in this research. This referential target is basically the same as the ones selected in the referenced studies [20–22,27] for the purpose of knowledge integration.

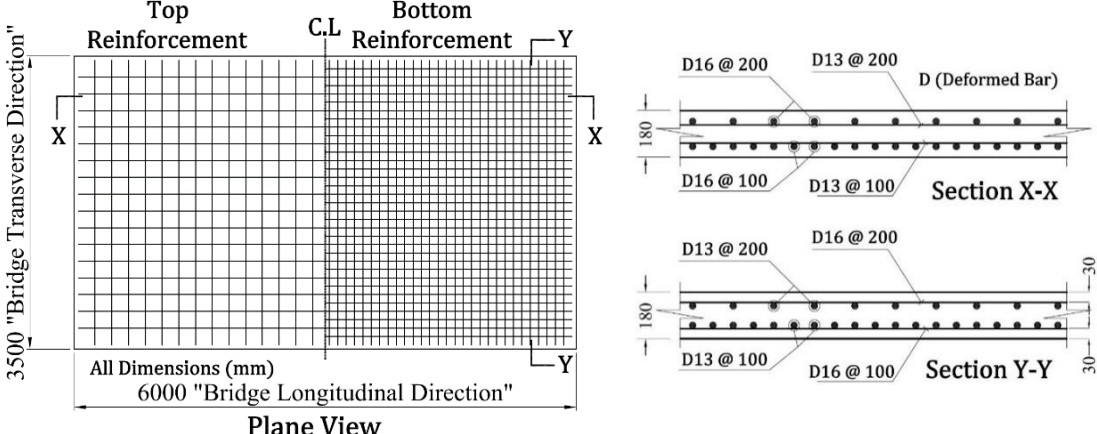

**Figure 5.** Dimensions and reinforcement arrangement of the studied slab.

### 2.2.2. Material Properties

The material properties of the concrete and steel of the referential RC slab are listed in Table 1 [20,21]. These values are followed by general designs used in past construction of highway bridge decks.

**Table 1.** Material properties of the referential slab.

| Material Type | | Concrete | Steel Reinforcement |
|---|---|---|---|
| Young's Modulus | N/mm$^2$ | 24,750 | 205,000 |
| Compressive Strength | N/mm$^2$ | 30 | 295 |
| Tensile Strength | N/mm$^2$ | 2.2 | 295 |
| Specific Weight | kN/m$^3$ | 24 | 78 |

### 2.2.3. Referential Loads & Boundary Conditions

On the basis of Japan's Specifications for Highway Bridges-Part III [35], we set up the deck to be subjected to a traveling wheel-type load of 98 kN, as shown in Figure 6. The speed of the running wheel for simulation was specified to be 60 km/h, which is the legal speed limit for national routes. The wheel load length and width used in the simulation were 500 mm and 250 mm, respectively, in reference to the width of vehicle tires, as shown in Figure 6. Hinged supports that allow free rotation were chosen, but the translation movement was restrained with reference to real conditions. Compared to real bridge conditions, this boundary may lead to a somewhat shorter fatigue life to be on the safe side. These are the same dimensions as those adopted by the past laboratory experiments against which the present fatigue simulation is examined and validated.

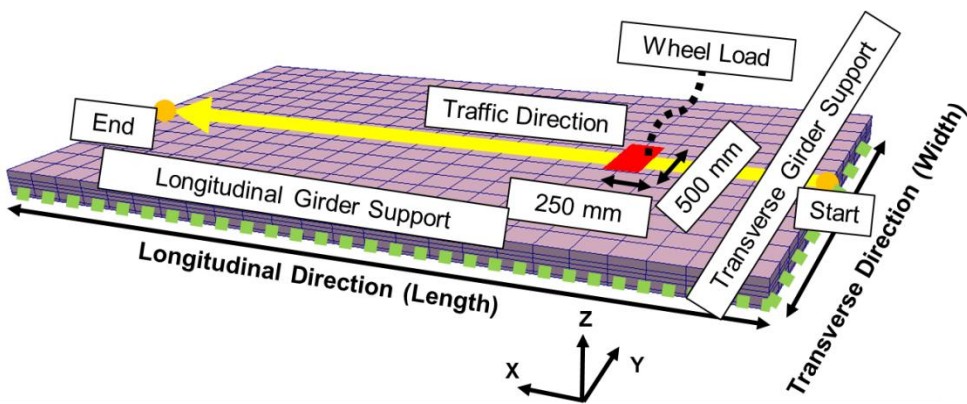

**Figure 6.** Wheel traveling load on the studied deck.

### 2.2.4. Failure Criteria

Based on the past experiments [25], the fatigue limit state was specified by the central live load deflection [36]. When the live load deflection defined by Equation (1) reaches the limit state deflection, which corresponds to no bond between the concrete and main reinforcement, it is judged that the slab comes up to the fatigue failure. It has been confirmed that the specific deflection associated with no bond is on the order of 2.5 times to 3.5 times its initial value [37], and the authors chose to apply the same criterion, denoted by Equation (2), as in past studies [20,21].

In fact, the fatigue life of RC decks depends on several aspects such as dimensions, material properties, reinforcement ratio, and load values. The limit state criterion was found to be rational as the properties of decks are included explicitly in the mean values of live load deflection. In almost all cases, out-of-plane shear failure of concrete occurs before the fatigue rupture of reinforcing bars [38]. In fact, no failure of reinforcement has been reported in past laboratory tests nor has been observed in real bridges. This can be attributed to the weakness of thinner concrete slabs rather than the fatigue strength of reinforcement. Thus, the limit state criterion is expected to be practical for the development and implementation of effective maintenance plans.

$$\delta_{L,N} = \delta_{1,N} - \delta_{2,N}, \tag{1}$$

$$\delta_{L,N} / \delta_{L,0} \geq 3.0, \tag{2}$$

where $\delta_{L,N}$ is the central live load deflection at N cycles, $\delta_{1,N}$ is the central total deflection at N cycles at the loading stage, $\delta_{2,N}$ is the central total deflection at N cycles at the unloading stage, $\delta_{L,0}$ is the initial live load deflection, and $N_f$ is the number of cycles corresponding to $\delta_{L,N}$ (Equation (2)).

### 2.2.5. Fatigue Life of Non-Damaged Deck

Figure 7 shows the relation of the wheel load cycles and the total central deflection for the referential deck (initially un-cracked), which is obtained by the multi-scale simulation program [20,21]. According to the failure criteria mentioned in Section 2.2.4, the fatigue life of this model is estimated at 221 million cycles. The live load deflection at the first cycle (A) is 1.4 mm, while that at failure (B) is 4.3 mm. Finally, the total deflection of 7.8 mm was specified as the value for failure for all the studied crack patterns.

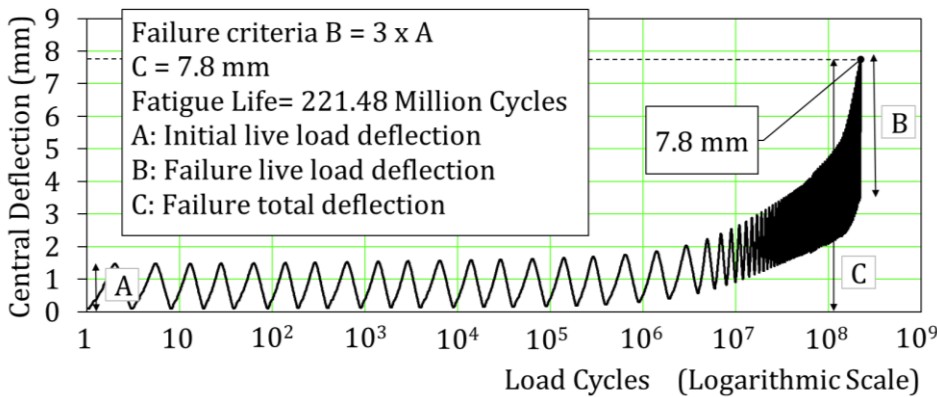

**Figure 7.** Load cycles versus central total deflection for the referential RC deck.

### 2.3. Randomized Artificial Crack Patterns (RACP)

To complement missing crack patterns over real decks, randomized artificial crack patterns are utilized. Figure 8 shows the basic RACP procedure concept. Here, the random variable is the number of elements (1–336). To avoid the overlap of case studies and reduce the effect of crack locations as much as possible, the randomization range of elements was limited to the first quarter of the deck.

The other quarters were provided with the same cracks obtained from the first quarter by using the laws of symmetry in both the longitudinal and transverse directions, as shown in Figure 9.

The crack angles were arranged in sequential order (each crack pattern has a single crack orientation) to cover all possible ranges from 0 degrees to 180 degrees with the step-interval of 15 degrees. The cracks with angles close to 0 degrees, 90 degrees, 180 degrees most extend the remaining fatigue life. Therefore, the smaller step interval of two degrees was further applied for the cases near 0 degrees, 90 degrees, and 180 degrees. The RACP produces continuous cracks, and three shapes of continuity are produced: x-direction, y-direction, or diagonal, on the basis of the sequential angle.

Both the crack width (0.3 mm) and crack density (average strain = 0.05%) were kept fixed in the RACP scheme to allow independent study of the effect of crack orientation, with the average strain parameter calculated by Equation (3). Although the RACP is based on randomization of elements, it can avoid repetition of cracking patterns. If a crack pattern is repeated, the RACP will continue randomization until a unique one is obtained.

It should be noted that the induced pre-crack is set to reach the upper mesh layer whose size is almost the same as that of the neutral axis of the studied RC deck [26].

$$\varepsilon_{\text{avg.}}(\text{A.S}) = \frac{\sum_{k=1}^{k=n}\left(\varepsilon_{xx} + \varepsilon_{yy}\right)}{n}, \tag{3}$$

where $\varepsilon_{\text{avg.}}$ is the average strain on the bottom surface of the RC deck, $\varepsilon_{xx}$ is the concrete's normal strain in x-direction for the kth element, $\varepsilon_{yy}$ is the one in y-direction, and n is the total number of elements (336 in this study).

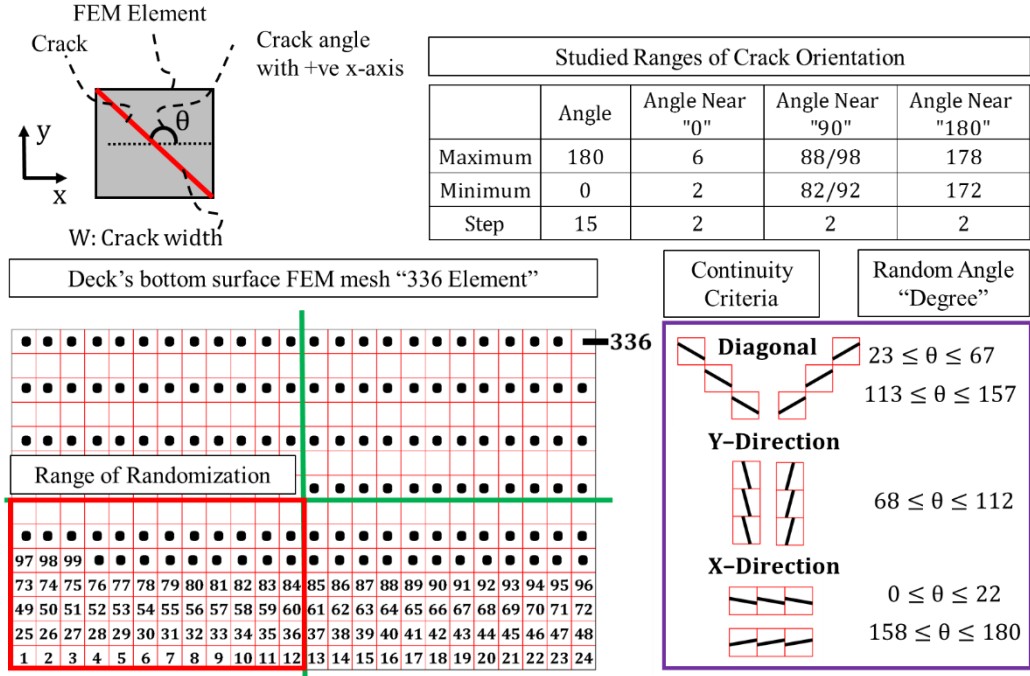

**Figure 8.** Randomized artificial crack pattern (RACP) scheme.

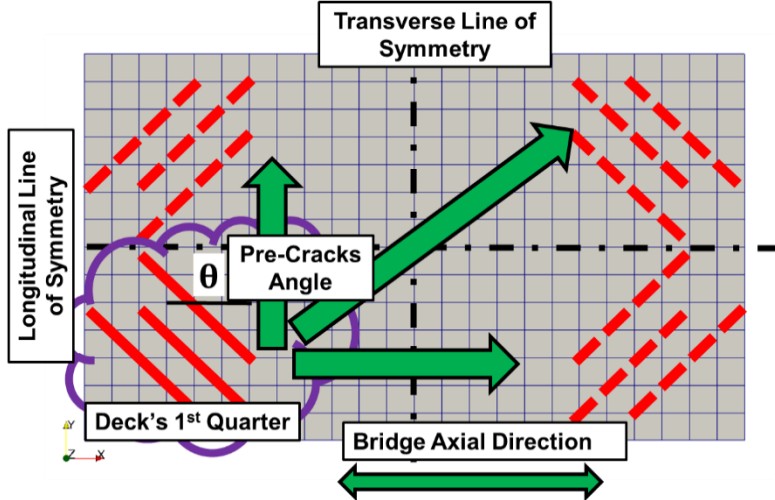

**Figure 9.** Utilization of laws of symmetry in RACP scheme.

## 3. Crack Patterns & Simulation Results

By utilizing the RACP, 148 crack patterns with a wide range of crack orientations (θ) were artificially produced. Figure 10 shows an example of the studied dataset. These crack patterns were analyzed for remaining fatigue life by the integrated multi-scale simulation program [23,24] and the pseudo-cracking method [25,26]. In spite of the fixed crack density, a wide range of fatigue life from 0.45 to 2.0 normalized by the non-cracked reference was obtained, as shown in Figure 11. The crack orientation was confirmed to be substantial.

Figure 12 shows the relation of the crack orientations and the remaining fatigue life of the crack patterns. It is clear that the cracks that are nearly parallel to the longitudinal or transverse direction (close to 0 degrees, 90 degrees, 180 degrees) most extend the remaining fatigue life, even beyond the undamaged sound condition. Here, the pre-cracks arrest the propagation of the post-shear cracks [20,21]. For other crack orientations, however, the remaining fatigue life remains almost unchanged and the effect of the crack orientations on the remaining fatigue life is moderate.

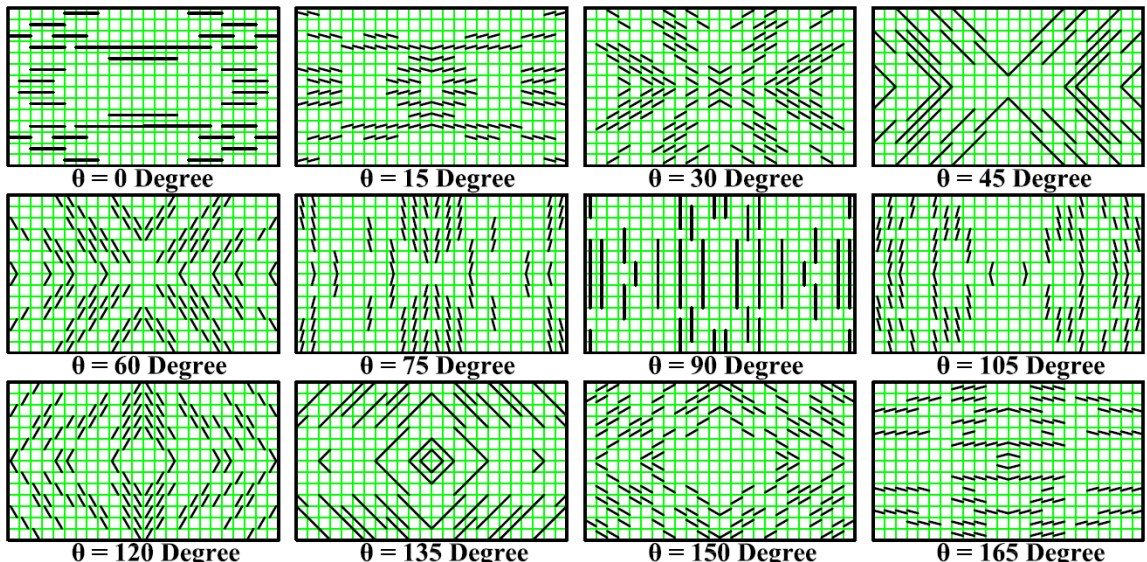

**Figure 10.** Example of the crack patterns for the studied dataset.

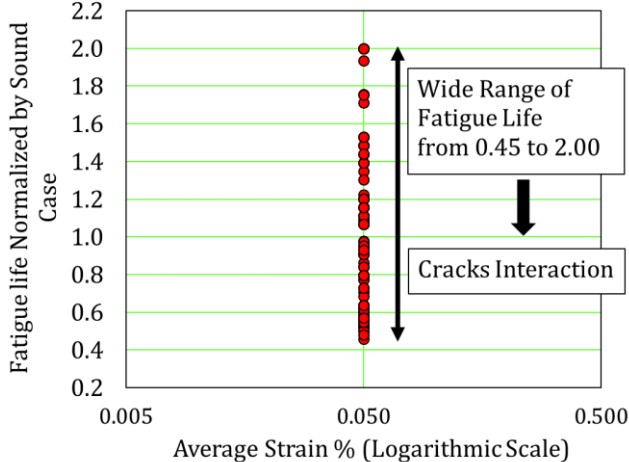

**Figure 11.** Relation of the average strain and the normalized fatigue life of the studied crack patterns.

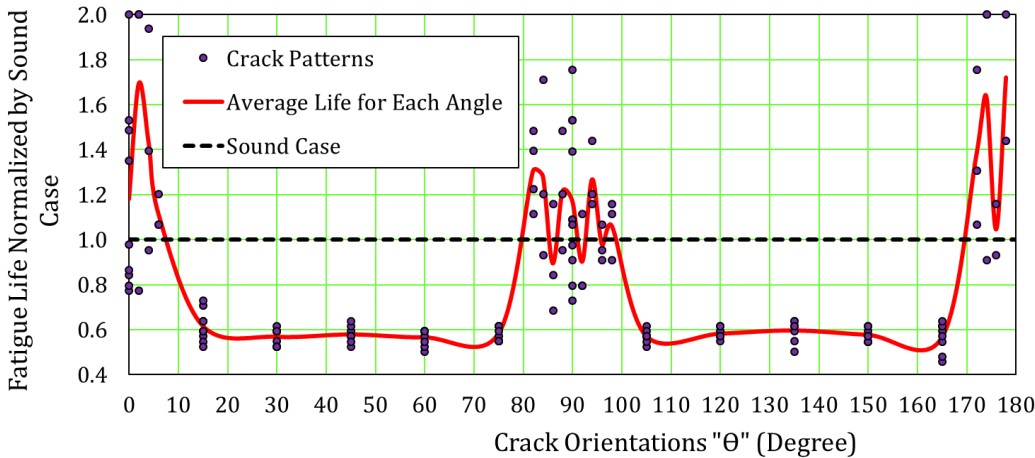

**Figure 12.** Relation of the crack orientations and the remaining fatigue life.

## 4. Discussions

### 4.1. Coupled Flexure-Shear Mode–Crack Arrest Mechanism

The crack arrest mechanism under shear was previously investigated for the case of RC beams [18] in which pre-cracks were introduced by flexural actions. Afterwards, the shear forces were applied to examine the propagation of diagonal shear cracks. It was reported that pre-cracks stop the propagation of post-shear diagonal cracks, leading to upgraded static capacity even greater than the sound condition of no pre-cracks. Once the stop mechanism arises, shear cracks are arrested by the preceding cracks and more energy is required to merge the post-shear cracks together with the preceding ones.

This phenomenon is highly dependent on the crack width, as shown in Figure 13. If pre-cracks hardly open, the diagonal shear crack can easily propagate across the pre-cracks. In this case, the shear stress is fairly transferred along crack planes by the aggregate interlock [18,20]. On the other hand, if the crack width of pre-cracks is larger, the local shear stress cannot be successfully transferred along pre-cracks' planes. This reduced shear stiffness along cracks causes stress release, which may not produce other diagonal cracks. Then, the crack arrest mechanism causes an elevated static capacity of RC members in shear. On that basis, RC decks also exhibit the crack arrest mechanism, and enhanced fatigue life can be realized as shown in Figure 12 [20,21].

The crack arrest mechanism in RC members is similar to the drilled stop-holes in steel members. It was reported that stop-holes stimulate the retardation of crack growth, where the crack tip is

transferred to notch and the stresses at the crack tip reduce significantly, followed by extended fatigue life [28–31]. Based on this analogy (see Figure 14), it can be said that the pre-cracks in concrete are like the stop-holes drilled through steel members.

It is clearly demonstrated in the previous section that the cracks that approximate the longitudinal and transverse directions of the deck most extend fatigue life. In consideration of the diagonal shear crack inside RC decks, the crack arrest mechanism is highly dependent on the crossing angle ($\phi$) of the pre-cracks and the post-shear cracks (see Figure 15). The crack arrest mechanism may be active when less shear stress is transferred across crack planes. In fact, it was demonstrated that the crack arrest mechanism comes up when pre-cracks and post-cracks intersect non-orthogonally [18]. Figure 16 shows the horizontal projections of pre-cracks and post-shear cracks in the x-y plane of the RC deck, where the post-shear cracks should mechanically be nearly diagonal in the horizontal projection.

If the pre-cracks are nearly diagonal, the intersection of the post-shear cracks is either orthogonal or nearly parallel. Therefore, the crack arrest mechanism hardly acts. In the case of the parallel intersection, the pre- and post-cracks may divert from each other without interaction. However, if the pre-cracks approximate the transverse or longitudinal direction of the RC deck, the non-orthogonality of the cracks' intersection is satisfied, and the crack arrest mechanism works, leading to extended fatigue life [18].

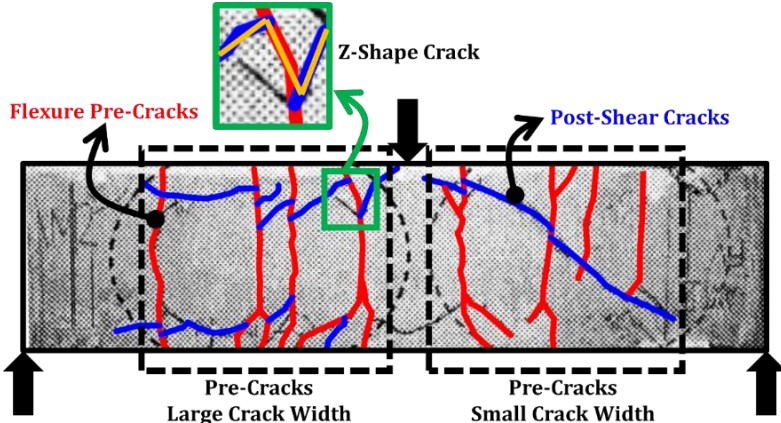

**Figure 13.** Pre-cracks stopping mechanism for RC beams.

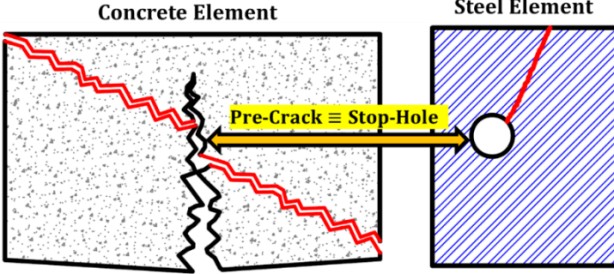

**Figure 14.** Stopping mechanism of crack growth in concrete and steel elements.

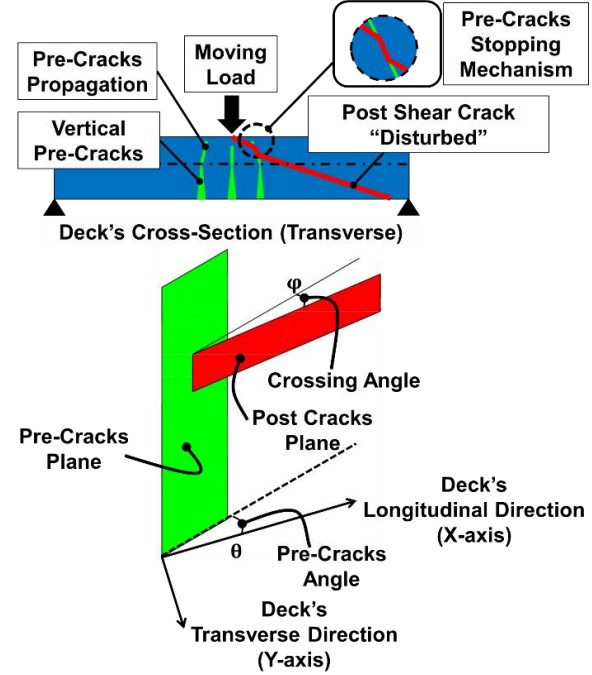

**Figure 15.** Interaction of pre-cracks and post-shear cracks.

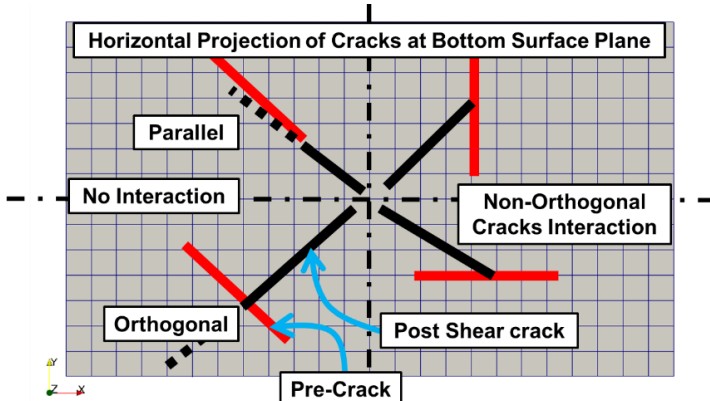

**Figure 16.** Horizontal projection of pre- and post-shear cracks at bottom surface of RC deck.

### 4.2. Effect of Cracks Location on Coupled Flexure-Shear Mode

Studied in this section is the effect of cracks location on the remaining fatigue life associated with cracks parallel to the transverse direction of the deck ($\theta = 90°$, see Figure 17). Here, we have a wide range of fatigue lives from 0.73 to 1.75 despite the same global crack density. It is found that there is a clear correlation between the average strain of the cracks in the central zone of the loading path (A.S*, see Figure 18) and the remaining fatigue life, as shown in Figure 19. When cracks are not present in the central zone (A.S* = 0), the crack arrest mechanism does not arise to enhance fatigue life.

On the other hand, by the increase in A.S*, the remaining fatigue life is extended, and the crack arrest mechanism becomes more effective. If the cracks pre-exist in the central zone, arrested cracks become dormant since the diagonal shear cracks start from the loading point, as shown in Figure 20.

When cracks exist outside the central zone of the wheel-loading path, the crack arrest mechanism is put off and the RC deck starts losing global stiffness prior to the limit for fatigue failure. For further investigation of this phenomenon, two crack patterns were focused on to obtain their static capacity: crack pattern (6) with (A.S*) equal to 0%, and crack pattern (11) with maximum (A.S*) equal to 0.084%. Figure 21 shows the relation of the deck's central deflection and the static load, where crack pattern

(11) has higher static capacity than that of the crack pattern (6) with about 14% despite their similar initial stiffness. The simulation results demonstrate that the crack arrest mechanism is effective when pre-cracks exist in the central zone.

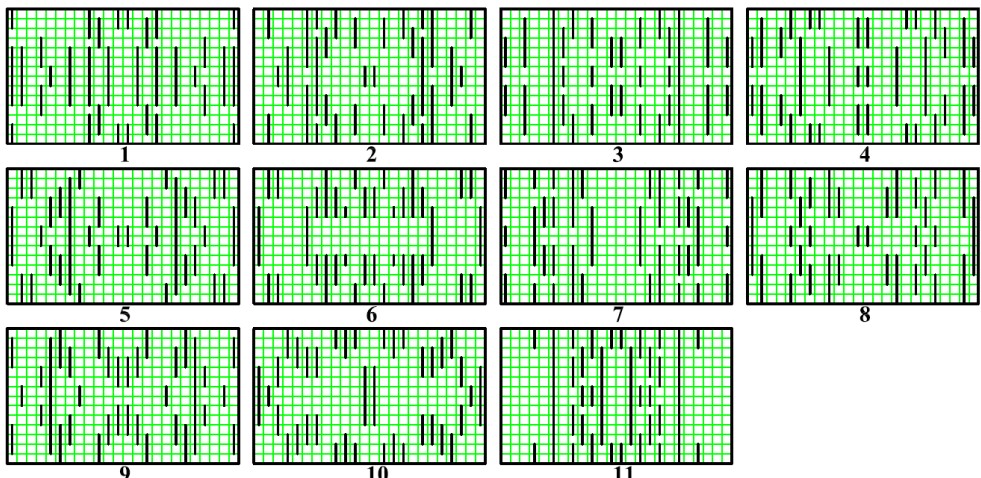

**Figure 17.** Artificial crack patterns with θ (90°).

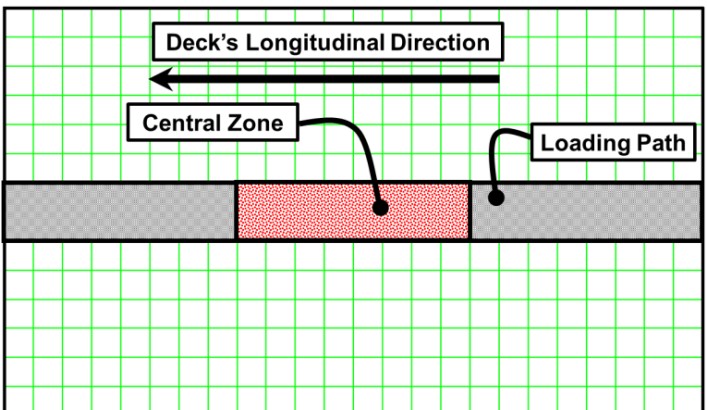

**Figure 18.** Finite element mesh of the bottom surface of the deck.

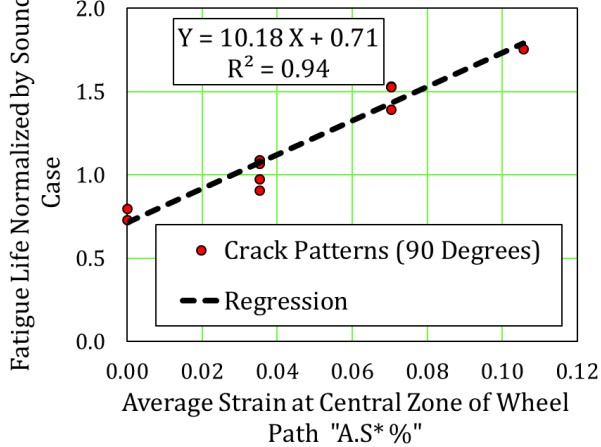

**Figure 19.** Relation of the average strain of the central zone of the loading path and the remaining fatigue life.

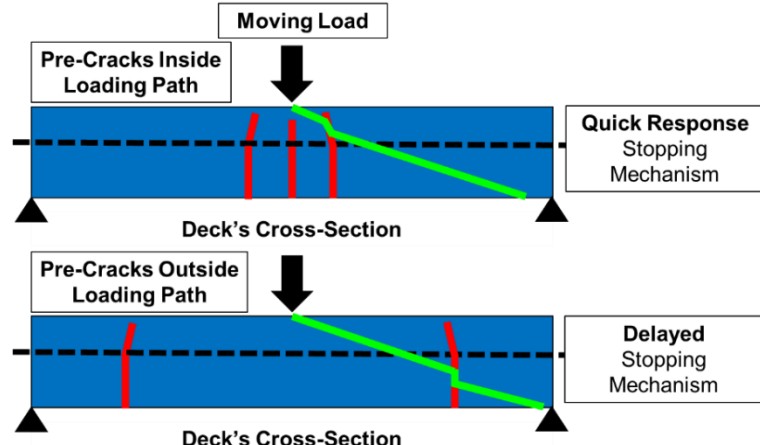

**Figure 20.** Pre-cracks stopping mechanism inside and outside loading path.

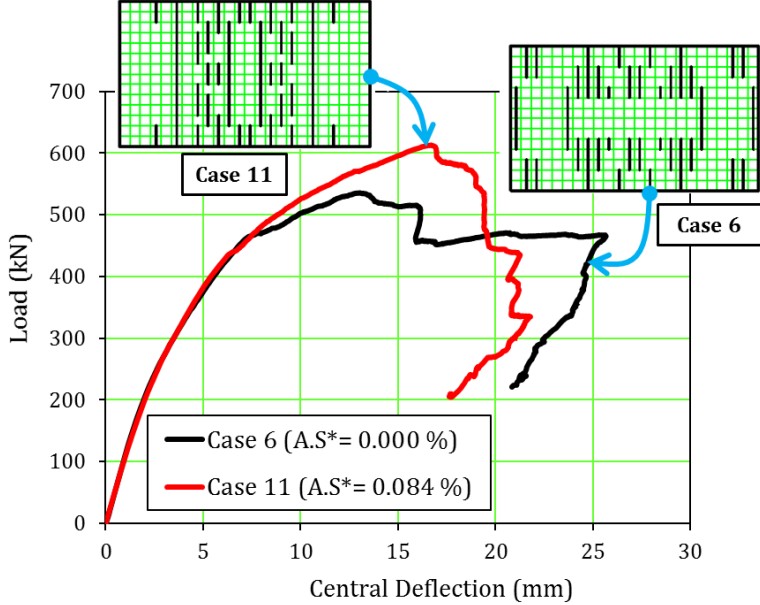

**Figure 21.** Central deflection of the RC deck and the static load for crack patterns (6 and 11).

### 4.3. High Risk Location of Cracking

In the previous research [21], a hazard map was achieved for the high-risk crack locations, i.e., the central zone and the corners of the RC deck, as shown in Figure 22, and highly coupled flexure and shear action was not explicitly included for safer and conservative assessment of remaining fatigue life. Thus, taking into consideration the uncertainty of cracks' depth on site, the depth of the cracks was previously intended to be shallow (non-penetrated cracks).

In this study, we targeted RC decks with penetrated cracks that are close to their top fibers in order to check different scenarios that may occur on site. In Section 4.2, we demonstrated that the central zone of the loading path is the location where the crack arrest mechanism is activated, and the fatigue life is elevated. This is beneficial in view of sustainable maintenance. Although the central zone of RC decks shall be designated as a higher-risk cracking region [21], the pre-crack arrest mechanism can be expected to lead to enhanced fatigue life.

On the other hand, the corners are also considered as higher-risk cracking locations, and the crack arrest mechanism hardly works there, as shown in Figure 22. Thus, cracks in the corners of RC decks are riskier than cracks in the central zone, where their depth is close to the top fibers of the RC deck. By combining the findings of the previous research [21] and those of this study, the upshot is that more

attention should be given to cracks at the corners than in the central zone of the RC deck during a bridge inspection.

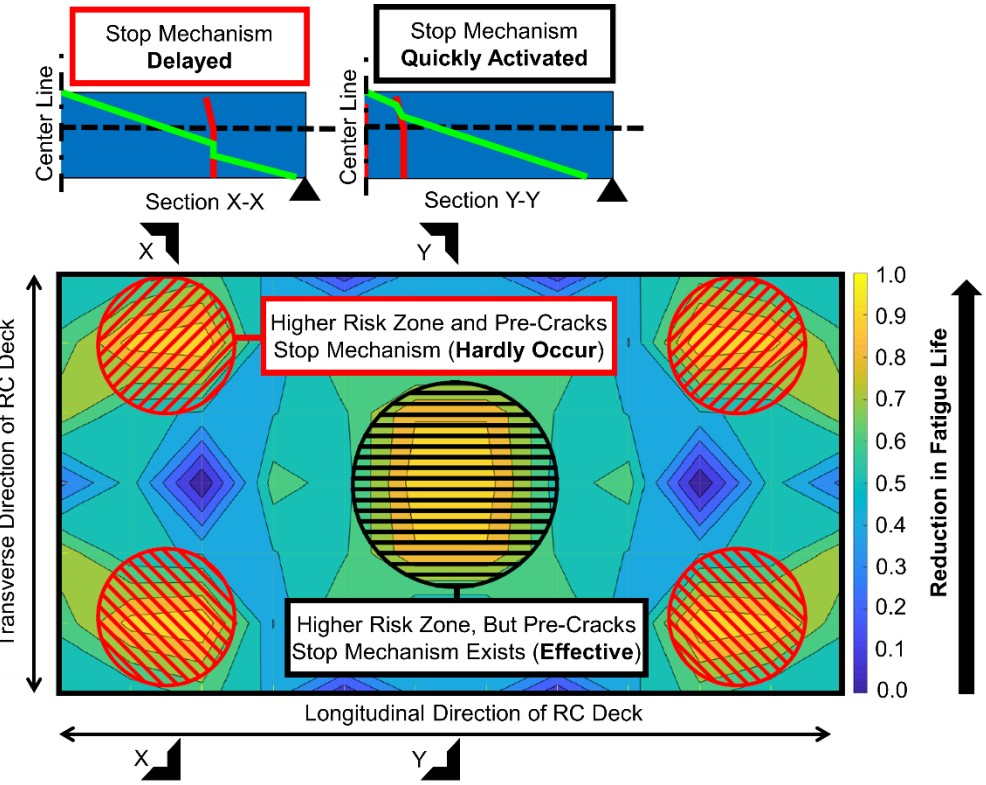

**Figure 22.** Hazard map for the locations of higher risk cracking.

## 5. Conclusions

The effect of crack orientation on the fatigue life of RC decks is investigated by using the multi-scale simulation and the pseudo-cracking method with respect to the most sensitive crack direction. On the basis of large numbers of reviewed crack patterns, the following conclusions are drawn.

i.      The crack orientation is highly associated with the coupled flexure-shear mode of failure, where pre-cracks tend to arrest the ensuing propagation of post-shear cracks. As a result, enhanced fatigue life is obtained.

ii.     The pre-crack arrest mechanism enables bridge maintenance managers to conduct a fair assessment of RC decks that present a rough appearance but have mechanically enhanced fatigue life.

iii.    The cracks that approximate the deck's longitudinal or transverse direction are understood to most extend the remaining fatigue life of RC decks.

iv.     The central zone of the loading path is found to be the location where the crack arrest mechanism is effective, and the remaining fatigue life is elevated.

v.      During inspections, careful attention shall be paid to cracks at the corners of RC decks since the stopping mechanism that arrests diagonal cracking does not effectively function there.

**Author Contributions:** E.F. conducted and analyzed the numerical studied cases; Y.T. and K.M. supervised the analytical process and developed the simulation program; E.F. and K.M. wrote the paper.

**Funding:** This study was financially supported by the Council for Science, Technology, and Innovation, "Cross-ministerial Strategic Innovation Promotion Program (SIP), Infrastructure Maintenance, Renovation, and Management" granted by the Japan Science and Technology Agency (JST).

**Acknowledgments:** The authors extend their appreciation to Yozo Fujino of Yokohama National University and The University of Tokyo, for his valuable advice and encouragement to bridge multiple disciplines in engineering.

**Conflicts of Interest:** The authors declare no conflict of interest.

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
