# Peer review of "Effect of Crack Orientation on Fatigue Life of Reinforced Concrete Bridge Decks"

_applsci, doi:10.3390/app9081644_

Round 1

Reviewer 1 Report

Comments:

The abstract should contain the following information:

Objectives of the work.

Novelty of the work.

Methodology.

Results.

The authors have missed to write about the proposed methodology and the main conclusions from their research.

The literature search (introduction) should be improved. Works dealing with orientation-induced fatigue crack growth retardation (e.g. Pavlou D.G.) or stop hole techniques (e.g. Rege K.) should be cited.

The authors should provide a theoretical support about the effect of the crack orientation on fatigue crack propagation (e.g. with the principle of minimum potential energy).

I recommend the acceptance for publication under the condition that the authors will address the above comments.

Author Response

(1) The abstract is updated, where the methodology is clearly stated. In addition, the conclusions and findings are updated at the final part of the abstract as follows. This part is indicated in yellow in the revised manuscript.

“Abstract: In visual inspection of bridges at sites, much attention is given to the density and width of cracks of concrete, but little attention is paid to crack orientation for the diagnosis of bridge performance. In this research, the effect of crack orientation on remaining fatigue life of reinforced concrete (RC) bridge decks is investigated for crack patterns with a wide range of possible crack orientations. The data assimilation technology of multi-scale simulation and the pseudo-cracking method, which are widely validated for fatigue-lifetime simulation, are utilized in this study. The impact of the crack direction on fatigue life is found to be associated with the coupled flexure-shear mode of failure, and the mechanism to arrest shear cracking by preceding cracks is quantitatively estimated. This mechanism is similar to the stop-hole to prevent fatigue cracks in steel structures and it enables us to enhance the fatigue life of RC decks. It is demonstrated that the crack orientations that approximate the longitudinal and transverse directions of RC decks are the ones that most extend remaining fatigue life. Finally, the higher risk cracking locations on the bottom surface of RC decks are discussed, presenting information of use to site inspectors.”

(2) The introductory section is updated with one paragraph to discuss stop-hole drilled in steel members and its similarity (analogy) to the pre-cracks arrest mechanism in RC members. Reference is added [31] together with three references [28,29,30] of the stop-hole in steel members, which were added before in the first submission. And the following paragraph of Section 4.1 is strengthened as;

“The crack arrest mechanism in RC members is similar to the drilled stop-holes in steel members. It was reported that stop-holes stimulate the retardation of crack growth, where the crack tip is transferred to notch and the stresses at the crack tip reduce significantly, followed by extended fatigue life [28-31]. Based upon this analogy (see Figure 13), it can be said that the pre-cracks in concrete are like the stop-holes drilled through steel members.”

 [31] Mikkelsen, O.; Rege, K.; Hemmingsen, T.; Pavion, D.G. Numerical estimation of the stop holes-induced fatigue crack growth retardation in offshore structures taking into account the corrosion effect. In The 27th International Ocean and Polar Engineering Conference. International Society of Offshore and Polar Engineers, San Francisco, California, USA, 25-30 June 2017.

(3) The theoretical background of the cracks interaction is updated in Section 2.1, where they are generally the constitutive laws of concrete integrated in the multi-scale simulation program. Added paragraph is marked in yellow.

Reviewer 2 Report

The manuscript describes the discoveries on the effect of crack orientation on the fatigue life of reinforced concrete (RC) decks. The study expands on analytical analysis based on behavior models developed and validated by the authors in previous publications, which are appropriately referenced in the manuscript.

The authors describe how crack density and width are usually taken into consideration when performing fatigue life analysis on reinforced concrete decks, however, crack orientation in such decks are not commonly assessed in previous studies.

The authors utilized randomized artificial crack patters to develop cracks on a RC deck from a typical bridge deck supported by steel-girder (representative from authors location). The cracks were generated at the first quarter of the deck and symmetry was then applied in the remaining quarters. The methodology for generating randomized cracks is properly described in the manuscript and it seems adequate for the analysis. This methodology allowed the authors to generate a variation of the crack patterns (referred into angles) for the studied dataset.

The authors previously investigated crack arrest mechanism under shear for the case of RC beams, where they introduced pre-cracks by flexural actions. They used the validated model in this study to determine the effect of crack orientation by keeping crack width and crack density constant.

The authors found a dependency of crack orientation with couple flexure-shear model. They also found that parallel and transverse cracks to be the most sensitive to fatigue life. Effects in central zone loading and recommendation on inspections are also reflected in their conclusions. The methodology described in the manuscript seems appropriate to the analysis and the conclusions are properly obtained from the analysis of the results. I recommend the manuscript for publication.

Author Response

We appreciate the reviewer's comprehensive comments and understanding of the paper applied.

Reviewer 3 Report

The paper investigates the effect of crack orientation on fatigue life of RC bridge decks. The main question is related to the fact that all the considerations are based on numerical simulations without any reference to a real situation or experimental findings. From this point of view, the paper is then a numerical exercise making use of numerical tools for the simulation of the fatigue behavior. Additionally, the paper is not well written since it is unclear how numerical simulations were really performed and the conclusions are trivial. As a conclusion, the paper cannot be accepted for publication

Author Response

(1) The findings of this research are on the basis of simulation results, as the reviewer kindly mentioned. Here, the data assimilation technology of the integrated system of the multi-scale simulation and the pseudo cracking method have been widely validated for fatigue lifetime simulation of RC decks based upon their site-inspected cracks. Then, the discussion of this data assimilation technology (methodology of this research) is briefly added with its theoretical background in Section 2.1. In addition, two examples of the validations for realistic RC decks and experimental specimens are also discussed as a chief reference in the volume of this journal, which were conducted in previous studies in the authors’ team.

The added part is indicated in yellow.

(2) The main highlight of this research is the pre-cracks arrest mechanism for the post shear cracks (couple flexure-shear mode), which enables us to upgrade the fatigue life of RC decks and to conduct fair assessment for RC decks that look rough but have longer life mechanically. This main highlight is added to the conclusion section as,

“The pre-crack arrest mechanism enables bridge maintenance managers to conduct a fair assessment of RC decks that present a rough appearance but have mechanically enhanced fatigue life.”

This part is indicated in yellow.

Round 2

Reviewer 1 Report

The title of the manuscript is "Effect of Crack Orientation on Fatigue Life of Reinforced Concrete Bridge Decks". Apart from the mapping system of the cracks, the authors should focus on the Physics and the physical principles of the effect of crack orientation on fatigue life. The authors failed to address the comment "The authors should provide a theoretical support about the effect of the crack orientation on fatigue crack propagation (e.g. with the principle of minimum potential energy)". I recommend them to read some relevant papers, e.g.:

Sih G.C., Barthelemy B.M., Mixed mode fatigue crack growth predictions, Engineering Fracture Mechanics, 13(3), 1980, 439-451.

Pavlou D.G., Labeas G.N., Vlachakis N.V., Pavlou F.G., Fatigue crack propagation trajectories under mixed-mode cyclic loading, Engineering structures, 25, 2003, 869-875.

and improve the manuscript accordingly. They should also check the list of references. Some author names are wrong (e.g. in ref. 30).

Author Response

(1) The summary of the basis on the propagating crack is added in Section 2.1 as below and three references [32,33,34] is added to explain the characteristics of concrete fracture as non-metaric cement composite.

 "As concrete is a cementitious composite, a single crack opening (Mode-I) though cement paste solid exhibits tension softening and the fracture energy is much less than that of crack shear transfer of Mode-II (see Figure 2) owing to interlocking of rough crack surfaces. Then, blunt multiple cracks are consequently dispersed and the direction of subsequently propagating cracks is assumed to coincide with that of the updated principal stress, which satisfy the equilibrium with non-orthogonal preceding cracks. It means the minimum fracture energy of the whole analysis domain associated with the shear transfer along crack planes and tension softening normal to cracks [25]. The high shear transfer along preceding crack planes of concrete is a fatigue resistant mechanism which differs from the case of smooth preceding crack planes without shear transfer [32].

     Generally, existing crack will propagate if its propagation is associated with a decrease in the total energy of the system [33,34]. In the case of RC members, the post-shear cracks propagate through the preceding cracks when the shear can transfer from one crack’s plane to another. Therefore, the system may easily reach the point of minimum potential energy. However, when the preceding crack’s arrest mechanism occurs, the system is in stable state, where more energy is required to create independent shear cracks until forming failure path. This is so called cracks interactions, which is explained in details in later sections."

(2) We re-cheked the references and revised including the format and type of letters.

Reviewer 3 Report

According to the the reqiests of this reviewer, the author added wo examples of the validations for realistic RC decks and experimental specimens are also discussed. Also the Discussions and Refernces sessions were improved and then the paper can be accepted for pubblication in the present form

Author Response

We appreciate the reviewer's suggestions and guidance to improve the manuscript.

Round 3

Reviewer 1 Report

In my review I recommended the authors should check the list of references. Some author names are still wrong, e.g. in ref. 30 the correct is "Pavlou DG" instead of "Pavion DG".

Author Response

In my review I recommended the authors should check the list of references. Some author names are still wrong, e.g. in ref. 30 the correct is "Pavlou DG" instead of "Pavion DG".

We appreciate your careful check. Reference no. 30 is updated as follows.

30.         Mikkelsen, O.; Rege, K.; Hemmingsen, T.; Pavlou, D.G. Numerical estimation of the stop holes-induced fatigue crack growth retardation in offshore structures taking into account the corrosion effect. In The 27th International Ocean and Polar Engineering Conference. International Society of Offshore and Polar Engineers, San Francisco, California, USA, June 2017, 25-30.